# Breastfeeding Practices and Postpartum Weight Retention in an Asian Cohort

**DOI:** 10.3390/nu16132172

**Published:** 2024-07-08

**Authors:** See Ling Loy, Hiu Gwan Chan, Joyce Xinyun Teo, Mei Chien Chua, Oh Moh Chay, Kee Chong Ng

**Affiliations:** 1Department of Reproductive Medicine, KK Women’s and Children’s Hospital, 100 Bukit Timah Road, Singapore 229899, Singapore; 2Duke NUS Medical School, 8 College Road, Singapore 169857, Singapore; chua.mei.chien@singhealth.com.sg (M.C.C.); chay.oh.moh@singhealth.com.sg (O.M.C.); ng.kee.chong@singhealth.com.sg (K.C.N.); 3Department of Pediatric Endocrinology Service, KK Women’s and Children’s Hospital, 100 Bukit Timah Road, Singapore 229899, Singapore; chan.hiu.gwan@kkh.com.sg; 4Division of Medicine, KK Women’s and Children’s Hospital, 100 Bukit Timah Road, Singapore 229899, Singapore; joyce.teo.xy@kkh.com.sg; 5Department of Neonatology, KK Women’s and Children’s Hospital, 100 Bukit Timah Road, Singapore 229899, Singapore; 6Department of Pediatrics, KK Women’s and Children’s Hospital, 100 Bukit Timah Road, Singapore 229899, Singapore; 7Yong Loo Lin School of Medicine, National University of Singapore, National University Health System, Singapore 119228, Singapore; 8Lee Kong Chian School of Medicine, Nanyang Technological University, 11 Mandalay Road, Singapore 308232, Singapore; 9Executive Office, Changi General Hospital, 2 Simei St. 3, Singapore 529889, Singapore

**Keywords:** breastfeeding, maternal obesity, postpartum weight retention, Singapore

## Abstract

This study examines relationships between breastfeeding practices and postpartum weight retention (PPWR) at 6 and 12 months postpartum among 379 first-time mothers participating in a clinical trial in Singapore. We categorized feeding modes at 6 months into exclusive breastfeeding, mixed feeding, and exclusive formula feeding. Participants were analyzed in two groups based on their PPWR assessment at 6 and 12 months postpartum, with complete datasets available for each assessment. We calculated PPWR by subtracting pre-pregnancy weight from self-reported weight at 6 and 12 months postpartum, defining substantial PPWR as ≥5 kg retention. Modified Poisson regression models adjusted for potential confounders were performed. At 6 and 12 months, 35% (*n* = 132/379) and 31% (*n* = 109/347) of women experienced substantial PPWR, respectively. Compared to exclusive breastfeeding, mixed feeding (risk ratio 1.85; 95% confidence interval 1.15, 2.99) and exclusive formula feeding (2.11; 1.32, 3.28) were associated with a higher risk of substantial PPWR at 6 months. These associations were slightly attenuated at 12 months and appeared stronger in women with pre-pregnancy overweight or obesity. This study suggests that breastfeeding by 6 months postpartum may help mitigate PPWR, particularly with exclusive breastfeeding. It also draws attention to targeted interventions to promote breastfeeding among women with overweight or obesity.

## 1. Introduction

Postpartum weight retention (PPWR), which measures weight change from pre-pregnancy to a specified period postpartum (typically 6 months and beyond), is common among women and is associated with increased risks of overweight and obesity, and cardiometabolic diseases [1,2]. PPWR spans the interpregnancy interval and contributes to cumulative weight gain in subsequent pregnancies [3]. This not only heightened the risk of maternal obesity and related pregnancy complications [4] but also increased the risk of obesity in offspring, perpetuating a vicious cycle of obesity [5]. PPWR represents a significant public health concern amid the escalating global obesity rates.

While the benefits of breastfeeding for maternal health are well-established, the evidence linking it with PPWR remains ambiguous [6]. A comprehensive review of 50 studies yielded inconclusive results; 30 reported minimal or no association with PPWR, possibly due to breastfeeding being assessed for a duration of less than 3 months or at a low intensity [6,7]. Conversely, a meta-analysis of 14 cohort studies indicated that breastfeeding for at least 6 months was associated with lower PPWR during the first year postpartum compared to formula feeding [8]. Notably, many of these studies focused on Western populations, with only a few conducted in Asian settings, where most assessed weight retention only during the early postpartum period up to 6 months [2,9,10].

In this study, we examine the association between breastfeeding practices recorded at 6 months postpartum and PPWR assessed at both 6 and 12 months postpartum, using data from the Community-enabled Readiness for first 1000 Days Learning Ecosystem (CRADLE) trial performed in Singapore. The assessment of breastfeeding practices was specifically focused at 6 months postpartum and not beyond, as this period is less influenced by weaning practices and has been suggested to significantly influence maternal weight outcomes [8]. We hypothesized that, compared to women practicing exclusive breastfeeding at 6 months, those practicing mixed or formula feeding would exhibit greater PPWR. We anticipated that these findings would provide supportive evidence for designing future interventions aimed at improving postpartum maternal metabolic recovery and weight management alongside promoting breastfeeding.

## 2. Materials and Methods

We drew data from the CRADLE, a three-arm randomized controlled trial that was designed to enhance parenting self-efficacy through behavioral nudges (Facebook group) and midwife-led continuity care (midwives group) during the first 1000 days for first-time parents. Participants from the Facebook group received weekly mobile messages containing information related to antenatal and postnatal care. Those from the midwife-led group received counseling and advice from midwives at various time points up to 6 months postpartum. The study was conducted according to the guidelines of the Declaration of Helsinki, approved by the Centralized Institutional Review Board of SingHealth (reference 2019/2781, 21 November 2019) and registered at ClinicalTrials.gov NCT04275765. The trial details were published previously [11]. Briefly, CRADLE targeted first-time pregnant mothers planning to reside in Singapore for the next 3 years, who were at least 17 years old and had internet access. The study excluded women with pre-existing chronic conditions (i.e., diabetes, mental illnesses, systemic lupus erythematosus, renal disease, cervical incompetence, and uterine malformation), multiple pregnancies, complicated pregnancies with fetal abnormality or those unable to understand English. Recruitment took place at the antenatal clinics of KK Women’s and Children’s Hospital, Singapore, from June 2020 to December 2021.

### 2.1. Study Procedure

Participants provided sociodemographic characteristics and pre-pregnancy weight and height data at the recruitment visit. At the 6- and 12-month postpartum visits, participants were asked to record their breastfeeding practices and weight data through an online questionnaire.

#### 2.1.1. Assessment of Breastfeeding and Weight Measurement

Breastfeeding practice was assessed based on feeding mode over the past 7 days, with options including ‘only breast milk’, ‘a combination of breast milk, formula and/or water’, and ‘only formula, water, or other liquids but not breast milk’ [12]. Participants self-reported their weight (kg) and height (cm). PPWRs at 6 and 12 months were computed by subtracting pre-pregnancy weight from the weight at 6 and 12 months postpartum, respectively. Substantial PPWR was defined as weight retention of 5 kg or more, while normal PPWR was characterized by weight retention of less than 5 kg. This threshold, consistently adopted in our previous studies [13,14] and others [15,16], is associated with various health risks and risk factors, guiding targeted interventions.

#### 2.1.2. Assessment of Potential Confounders

We conducted a literature review and used a directed acyclic graph to identify potential confounders, focusing on common factors influencing both breastfeeding practices and PPWR. We identified the following confounders: maternal age, pre-pregnancy body mass index (BMI), ethnicity, education level, marital status, employment status, and total monthly household income [17,18,19,20]. We further adjusted for the intervention groups, defined as control, Facebook, and midwives, to reduce bias from the CRADLE intervention effect [11]. Maternal age, collected at the recruitment visit, was analyzed as a continuous variable. Pre-pregnancy BMI was computed using the formula weight(kg)/height(m)^2^ and also treated as a continuous variable. We categorized ethnicity into Chinese, Malay, Indian, and Others, with the latter including Arab, Burmese, Caucasian, Eurasian, Filipino, Indonesian, Italian, Japanese, Korean, and Vietnamese. Education level was categorized into post-secondary and below and university or above. Marital status was categorized as married and single/divorced. We categorized total monthly household income in Singapore dollars into three groups: S$5000 and below, S$5000–8000, and S$8000 and above based on the distribution of the total monthly income data.

### 2.2. Statistical Analysis

We presented categorical variables as frequencies and percentages and continuous variables as means ± standard deviations (SDs). We conducted the analysis by dividing participants into two groups based on their PPWR at 6 and 12 months postpartum, ensuring each group had the maximum number of complete datasets for both feeding practices and PPWR assessments. Participant characteristics by PPWR status at 6 and 12 months postpartum were compared using Pearson’s chi-squared test or Fisher’s exact test for categorical variables where applicable and independent *t*-tests for continuous variables. We employed multiple linear regression models and modified Poisson regression models to examine the relationship between feeding practices and PPWR in continuous and categorical forms, respectively, adjusting for potential confounders. We imputed the missing data for the confounding variables, including marital status at 6 months (*n* = 3) and 12 months postpartum (*n* = 2), using the most frequently occurring marital status at the respective time points. Results were reported as β-coefficients and risk ratios (RRs), each with their respective 95% confidence intervals (CIs). Considering the potential influence of BMI on the breastfeeding–PPWR association [8,21], we conducted an interaction test between feeding practices and pre-pregnancy BMI (categorized into <23 and ≥23 kg/m^2^ based on the Asian cut-off [22]) on PPWR (continuous) at 12 months postpartum. Guided by the *p*-for-interaction at 0.091, we stratified the analysis by pre-pregnancy BMI status. We also performed interaction tests between feeding practices and other potential confounders, all with *p*-for-interactions > 0.10. We conducted a sensitivity analysis to examine the robustness of the results by including only participants who had complete PPWR data at both 6 and 12 months postpartum (*n* = 332). All analyses were performed using Stata 18 (Stata Corporation).

## 3. Results

We recruited 548 participants at baseline, of whom 396 had both pre-pregnancy weight data and 6 months postpartum feeding practices data. Of these 396 participants, 16 had missing weight data at 6 months and 1 had an implausible PPWR at 6 months, resulting in 379 participants with complete and usable data at 6 months postpartum. At 12 months postpartum, 49 participants had missing weight data, leaving 347 participants with complete data. In total, 332 participants had complete baseline sociodemographic, feeding practices, and weight data at both 6 and 12 months postpartum (Figure 1). 

Appendix A shows the comparison of characteristics between participants with complete (*n* = 332) and incomplete data (*n* = 216). Despite similar PPWRs at 6 and 12 months, as well as similar distributions of intervention group, marital status, employment status, and feeding practices at 6 months (all *p* > 0.05), participants with complete data were more likely to be older (mean age 31.2 years [SD 3.6] vs. 30.4 years [SD 3.9]), have lower BMI (22.5 kg/m^2^ [SD 4.0] vs. 24.3 [SD 5.3]), be Chinese (79.8% vs. 51.6%), have tertiary education and above (81.0% vs. 65.1%), and have a total monthly household income of S$8000 and above (47.3% vs. 34.4%) compared to those with incomplete data (all *p* < 0.05).

### 3.1. Participant Characteristics and Substantial PPWR

Table 1 presents the participant characteristics by PPWR status at 6 and 12 months postpartum, divided into two groups to maximize the completeness of datasets for both feeding practices and PPWR assessments. Overall, participants experienced an average weight gain of 3.0 kg (SD 5.1) at 6 months and 2.7 kg (SD 4.9) at 12 months postpartum. Approximately one-third of participants encountered substantial PPWR at both 6 months (34.8%) and 12 months (31.4%). Those with substantial PPWR retained an average of 8.4 kg (SD 3.5) at 6 months and 8.3 kg (SD 3.3) at 12 months postpartum, significantly higher than those with normal PPWR, who retained an average of 0.1 kg (SD 3.0) and 0.1 kg (SD 3.1), respectively. Despite similar ages, pre-pregnancy BMIs, intervention group distributions, and marital statuses (all *p* > 0.05), participants with substantial PPWR at 6 months were more likely to be Malay (14.4% vs. 8.5%) or Indian (13.6% vs. 3.2%), have education below tertiary level (25.8% vs. 15.4%), be unemployed (11.4% vs. 3.6%), have a total monthly household income of S$5000 and below (31.8% vs. 19.0%), and were less likely to exclusive breastfeed by 6 months (13.6% vs. 30.8%), compared to participants with normal PPWR (all *p* < 0.05). At 12 months postpartum, similar associations were found where participants with substantial PPWR were more likely to be of Malay or Indian ethnicity, have lower educational levels, be unemployed, have lower household incomes, and were less likely to exclusively breastfeed compared to those with normal PPWR (all *p* < 0.05).

### 3.2. Association between Feeding Practices and PPWR

At 6 months postpartum, there were 94 (24.8%) participants practicing exclusive breastfeeding, 141 (37.2%) practicing mixed feeding, and 144 (38.0%) practicing exclusive formula feeding. Table 2 shows the associations between feeding practices and PPWR. At 6 months postpartum, participants practicing mixed feeding and exclusive formula feeding had higher mean PPWRs of 2.37 kg (95% CI 1.08, 3.65) and 3.24 kg (1.96, 4.52), respectively, compared to those exclusive breastfeeding. These results persisted in models adjusted for potential confounders. At 12 months postpartum, adjusted models also showed higher PPWRs for mixed feeding (1.92 kg; 0.60, 3.24) and exclusive formula feeding (2.79 kg; 1.44, 4.13) compared to exclusive breastfeeding, though the associations were attenuated. 

When PPWR was treated as the binary outcome, mixed feeding (adjusted RR 1.85; 95% CI 1.15, 2.99) and exclusive formula feeding (2.11; 1.32, 3.28) were associated with higher risks of developing substantial PPWR at 6 months, compared to exclusive breastfeeding (Figure 2b). These associations were attenuated at 12 months, with participants practicing mixed feeding and exclusive formula feeding showing 46% (1.46; 0.89, 2.39) and 88% (1.88; 1.16, 3.04) higher risks of developing substantial PPWR at 12 months, respectively (Figure 2d).

Sensitivity analyses on participants with complete data revealed consistent trends. At 6 months postpartum, adjusted models showed higher mean PPWRs for mixed feeding (2.21 kg; 95% CI 0.81, 3.62) and exclusive formula feeding (3.21 kg; 1.78, 4.64) compared to exclusive breastfeeding. At 12 months postpartum, PPWRs were 1.67 kg (95% CI 0.30, 3.03) for mixed feeding and 2.62 kg (1.23, 4.01) for exclusive formula feeding. Binary outcome analyses also revealed increased risks for substantial PPWR. At 6 months postpartum, the adjusted RR was 1.72 (95% CI 1.06, 2.79) for mixed feeding and 1.90 (1.18, 3.07) for formula feeding. By 12 months, these risks remained at 1.32 (0.79, 2.17) for mixed feeding and 1.80 (1.11, 2.92) for exclusive formula feeding (Appendix A). 

A stratification analysis by pre-pregnancy BMI status revealed that at 12 months postpartum, participants practicing mixed feeding or exclusive formula feeding exhibited greater PPWR compared to those exclusive breastfeeding, particularly among those in the higher BMI category at ≥23 kg/m^2^ (Appendix A).

## 4. Discussion

This longitudinal study, utilizing data from the CRADLE trial, examined the association between breastfeeding practices recorded at 6 months postpartum and PPWR assessed at 6 and 12 months postpartum among women in Singapore. Overall, a much lower proportion of women practiced exclusive breastfeeding by 6 months postpartum compared to those who used mixed or formula feeding. We found that approximately one-third of the women experienced substantial PPWR at both time points, retaining at least 5 kg above their pre-pregnancy weight. Women who engaged in mixed or exclusive formula feeding consistently demonstrated higher average PPWRs and were at a higher risk of developing substantial PPWR compared to those who exclusively breastfed. The increase in PPWR was notably more pronounced among women who exclusively formula-fed than those who mixed-fed. These associations persisted even after adjusting for potential confounders, including sociodemographic factors, pre-pregnancy BMI, and intervention effects. Furthermore, subgroup analyses revealed that women with pre-pregnancy overweight or obesity (BMI ≥ 23 kg/m^2^) who practiced mixed or formula feeding exhibited greater PPWR compared to their lean counterparts (<23 kg/m^2^). 

In a recent secondary analysis of 6264 women in Singapore, it was noted that 34% experienced an increase of 1–3 kg/m^2^ in BMI during the first 2 years of the interpregnancy period [4]. Consistently, our current study reveals that approximately 31% of women exhibited substantial PPWR, with a minimum weight gain of 5 kg by 12 months postpartum. This increment corresponds to an estimated BMI gain of nearly 2 kg/m^2^, calculated based on the average height of 1.6 m among the cohort participants. These findings underscore the critical necessity for national strategies to enhance postnatal care services. Such services should incorporate targeted lifestyle interventions alongside proactive breastfeeding support, particularly within the first 2 years post-delivery, an optimal intervention window for returning to pre-pregnancy BMI levels [4,23]. Specifically, dietary-based interventions appeared to be the most effective in promoting weight loss during this period [24]. Implementing effective lifestyle modifications is pivotal not only for mitigating PPWR but also for improving subsequent perinatal outcomes and enhancing long-term health in women [25].

Our findings confirm a positive association between mixed or exclusive formula feeding at 6 months and increased PPWR up to 12 months. This aligns with results from a meta-analysis of cohort studies, which showed that breastfeeding for at least 6 months reduced PPWR until at least 48 weeks postpartum [8]. Our data suggest that exclusive formula feeding had a more pronounced negative impact on PPWR than mixed feeding, indicating that any level of breastfeeding helps mitigate PPWR, with the most significant benefits arising from exclusive breastfeeding. This is consistent with findings from large cohorts in Taiwan [21] and Japan [2], where exclusive breastfeeding for 6 months resulted in greater postpartum weight loss than mixed feeding. The notable increase in PPWR associated with formula feeding can be attributed to the absence of metabolic benefits from breastfeeding. These include enhanced calorie expenditure and increased prolactin secretion, which facilitates the mobilization of adipose tissue stores and inhibits lipogenesis [1]. 

The meta-analysis by Jiang et al. [8] showed that the association between breastfeeding and reduced PPWR was stronger in women with a normal pre-pregnancy BMI. However, studies by Waits et al. [21] and Yamamoto et al. [2] observed a greater reduction in PPWR among women who were obese before pregnancy. Our findings align with these latter studies, indicating that in Asian populations, the absence of exclusive breastfeeding is associated with higher PPWR among women who were overweight or obese at the onset of pregnancy. Although our results did not achieve statistical significance due to a small sample size after stratification, the trends observed support the notion that breastfeeding exerts a more substantial effect on reducing PPWR among women with obesity, particularly within an Asian context. These divergent outcomes suggest that responses to breastfeeding might vary according to differences in metabolic health status and body fat composition among different population groups [1]. This underscores the need for further research to investigate how these metabolic variations influence the effectiveness of breastfeeding on PPWR. Such insights could inform more personalized approaches to managing postpartum weight.

### Limitations

This study has a few limitations that warrant consideration. Firstly, the assessment of feeding mode was based solely on the type of milk given to infants over the past 7 days, which did not capture potential discontinuations of these practices beyond this period. Our findings on the prevalence of breastfeeding (62%) at 6 months postpartum were lower than those reported in a prospective birth cohort study conducted in Singapore during 2009–2010. That study observed a 71% rate of breastfeeding at 6 months postpartum [26]. Notably, our study showed an increase in exclusive breastfeeding rates from 11% in the earlier cohort to 25% in the current analysis. This improvement suggests a positive shift in exclusive breastfeeding practices over time, although the methodological constraints of our feeding mode assessment might have influenced the accuracy of these observations. Secondly, gestational weight gain (GWG) was not considered in our analysis as this data was not collected. However, previous research has indicated that the intention and initiation of breastfeeding were not associated with GWG [27,28]. Thirdly, participants self-reported their weight data, potentially leading to an overestimation of the effect size. This method is subject to recall bias, and although generally reliable, its accuracy decreased among women categorized as overweight or obese and those experiencing weight fluctuations, with few under-reporting their weight by more than 10% [29]. Nonetheless, the comparable proportions of women having substantial PPWR as reported in this study and as measured in a previous large local study [4] support the reliability of our data. Finally, 80% of the women included in the analysis had tertiary education or higher. This demographic profile could disproportionately represent more educated women, introducing a potential selection bias. However, we observed no interaction between breastfeeding practices and educational levels on PPWR, suggesting the potential generalizability of our findings. Comparisons with participants who had incomplete data revealed significant differences in baseline characteristics such as age, pre-pregnancy BMI, ethnicity, education, and income levels. We attempted to mitigate this bias by adjusting for these characteristics in our analysis, as they were also identified as potential confounders.

## 5. Conclusions

This study demonstrates that in Singapore, women who exclusively formula feed or use mixed feeding methods by 6 months postpartum are at a higher risk for substantial postpartum weight retention at both 6 and 12 months postpartum, compared to those who exclusively breastfeed. The variations in the association between breastfeeding and postpartum weight retention across different weight statuses highlight the need for tailored interventions that consider pre-pregnancy body mass index and the underlying factors driving these differences. This approach is essential for developing comprehensive healthcare strategies for managing postpartum weight retention effectively across diverse populations.

## Figures and Tables

**Figure 1 nutrients-16-02172-f001:**
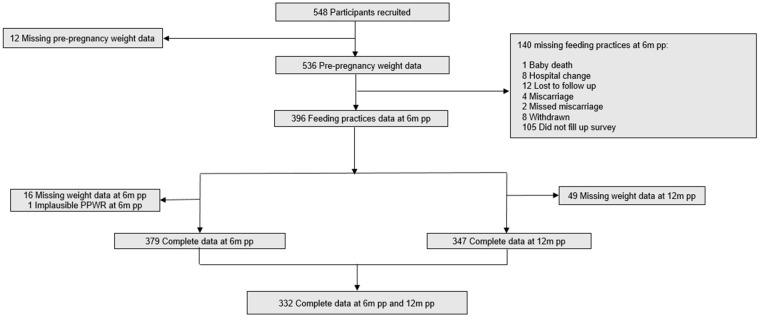
Flow chart for inclusion of participants in the study. M, months; pp—postpartum; PPWR—postpartum weight retention.

**Figure 2 nutrients-16-02172-f002:**
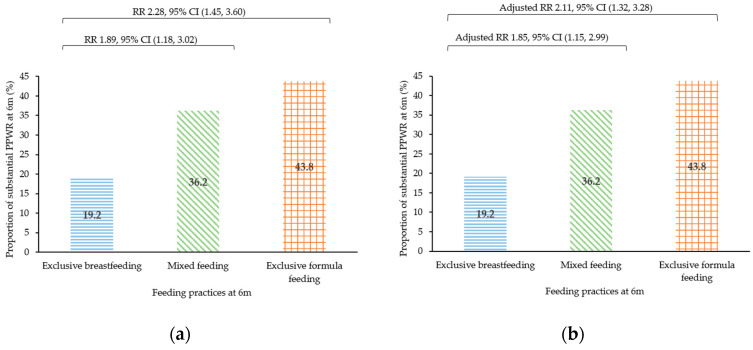
Association of feeding practices at 6 m pp and substantial PPWR (≥5 kg) at 6 m and 12 m: (**a**) Crude and (**b**) adjusted analysis of feeding practices at 6 m pp and substantial PPWR at 6 m; (**c**) Crude and (**d**) adjusted analysis of feeding practices at 6 m pp and substantial PPWR at 12 m. Models in (**b**,**d**) were adjusted for maternal age, pre-pregnancy body mass index, ethnicity, education level, marital status, employment status, total monthly household income, and intervention group. CI—confidence interval; m—months; pp—postpartum; PPWR—postpartum weight retention; RR—risk ratio.

**Table 1 nutrients-16-02172-t001:** Participants characteristics.

	PPWR at 6 m (*n* = 379)			PPWR at 12 m (*n* = 347)		
Characteristics	<5 kg (*n* = 247; 65.2%)	≥5 kg (*n* = 132; 34.8%)	*p* Value	<5 kg (*n* = 238; 68.6%)	≥5 kg (*n* = 109; 31.4%)	*p* Value
PPWR, kg, mean ± SD	0.1 ± 3.0	8.4 ± 3.5	<0.001	0.1 ± 3.1	8.3 ± 3.3	<0.001
Age at consent, year, mean ± SD	31.3 ± 3.6	31.0 ± 3.9	0.370	31.3 ± 3.5	30.9 ± 3.8	0.363
Pre-pregnancy BMI, kg/m^2^, mean ± SD	22.4 ± 4.3	23.0 ± 3.6	0.150	22.3 ± 4.1	22.8 ± 3.7	0.254
Intervention group, *n* (%)			0.823			0.151
Control	79 (32.0%)	46 (34.9%)		69 (29.0%)	43 (39.5%)	
Facebook	83 (33.6%)	44 (33.3%)		87 (36.6%)	35 (32.1%)	
Midwives	85 (34.4%)	42 (31.8%)		82 (34.5%)	31 (28.4%)	
Ethnicity, *n* (%)			<0.001			0.011
Chinese	208 (84.2%)	88 (66.7%)		199 (83.6%)	77 (70.6%)	
Malay	21 (8.5%)	19 (14.4%)		18 (7.6%)	15 (13.8%)	
Indian	8 (3.2%)	18 (13.6%)		9 (3.8%)	12 (11.0%)	
Others *	10 (4.1%)	7 (5.3%)		12 (5.0%)	5 (4.6%)	
Education Level, *n* (%)			0.014			0.015
Post-Secondary and below	38 (15.4%)	34 (25.8%)		37 (15.6%)	29 (26.6%)	
University and above	209 (84.6%)	98 (74.2%)		201 (84.5%)	80 (73.4%)	
Marital Status, *n* (%)			0.424			>0.950
Married	244 (98.8%)	129 (97.7%)		235 (98.7%)	108 (99.1%)	
Single/Divorced	3 (1.2%)	3 (2.3%)		3 (1.3%)	1 (0.9%)	
Employment Status, *n* (%)			0.003			0.002
Employed	238 (96.4%)	117 (88.6%)		230 (96.6%)	96 (88.1%)	
Unemployed/Student	9 (3.6%)	15 (11.4%)		8 (3.4%)	13 (11.9%)	
Total monthly household income, *n* (%)			0.013			0.033
S$5000 and below	47 (19.0%)	42 (31.8%)		48 (20.2%)	36 (33.0%)	
S$5000–8000	75 (30.4%)	39 (29.6%)		69 (29.0%)	28 (25.7%)	
S$8000 and above	125 (50.6%)	51 (38.6%)		121 (50.8%)	45 (41.3%)	
Feeding practices at 6 months pp, *n* (%)			0.003			0.005
Exclusive breastfeeding	76 (30.8%)	18 (13.6%)		71 (29.8%)	18 (16.5%)	
Mixed feeding	90 (36.4%)	51 (38.6%)		92 (38.7%)	39 (35.8%)	
Exclusive formula feeding	81 (32.8%)	63 (47.7%)		75 (31.5%)	52 (47.7%)	

BMI—body mass index; kg—kilograms; m—months; pp—postpartum; PPWR—postpartum weight retention; SD—standard deviation; S$—Singapore dollar. * Other ethnic groups include Arab, Burmese, Caucasian, Eurasian, Filipino, Indonesian, Italian, Japanese, Korean and Vietnamese. *P* values were calculated from Pearson’s chi-squared test or Fisher’s exact test where appropriate for categorical variables and independent *t* test for continuous variables.

**Table 2 nutrients-16-02172-t002:** Association between feeding practices and PPWR.

	PPWR at 6 m (*n* = 379)				PPWR at 12 m (*n* = 347)			
	Mean ± SD		β (95% CI)		Mean ± SD		β (95% CI)	
Feeding practices at 6 m pp	<5 kg	≥5 kg	Model 1	Model 2	<5 kg	≥5 kg	Model 1	Model 2
Exclusive breastfeeding	(−0.59) ± 2.96	7.26 ± 2.32	Reference	Reference	(−0.69) ± 3.06	7.43 ± 2.26	Reference	Reference
Mixed feeding	(−0.01) ± 2.91	9.08 ± 3.91	2.37 (1.08, 3.65)	2.23 (0.94, 3.52)	0.31 ± 3.15	8.67 ± 3.54	1.84 (0.53, 3.15)	1.92 (0.60, 3.24)
Exclusive formula feeding	0.99 ± 2.86	8.21 ± 3.39	3.24 (1.96, 4.52)	3.14 (1.83, 4.46)	0.57 ± 2.90	8.33 ± 3.47	2.79 (1.47, 4.11)	2.79 (1.44, 4.13)

β—regression coefficient; CI—confidence interval; kg—kilograms; m—months; pp—postpartum; PPWR—postpartum weight retention; SD—standard deviation. Model 1: crude model. Model 2: adjusted for maternal age, pre-pregnancy body mass index, ethnicity, education level, marital status, employment status, total monthly household income, and intervention group.

## Data Availability

Data described in the manuscript will be made available upon request pending approval by principal investigator (KCN) of the CRADLE study.

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
