# Peer review of "Breastfeeding Practices and Postpartum Weight Retention in an Asian Cohort"

_nutrients, 2024, doi:10.3390/nu16132172_

Round 1

Reviewer 1 Report

Comments and Suggestions for Authors

In the paragraph materials and methods the authors state that "The study excluded women with pre-74 existing chronic conditions", it would be better to specify which chronic conditions were excluded. Also, I suggest making the study limitations a subparagraph on its own in the discussion. 

Author Response

Reviewer 1
Thank you for the opportunity to revise our manuscript. We have made the necessary revisions, which are highlighted in yellow in the attached manuscript. Our point-by-point responses are provided below:

1.    Comment: In the paragraph materials and methods the authors state that "The study excluded women with pre-74 existing chronic conditions", it would be better to specify which chronic conditions were excluded. 

Response: We have specified the chronic conditions that served as exclusion criteria on page 2, line 76-80, as follows:
‘The study excluded women with pre-existing chronic conditions (i.e. diabetes, mental illnesses, systemic lupus erythematosus, renal disease, cervical incompetence, and uterine malformation), multiple pregnancies, complicated pregnancies with fetal abnormality or those unable to understand English.’

2.    Comment: Also, I suggest making the study limitations a subparagraph on its own in the discussion.

Response: We have added a subtitle named ‘Limitations’ a subparagraph within the ‘Discussion’ section, on page 8, line 284.  

Reviewer 2 Report

Comments and Suggestions for Authors

I read the manuscript entitled 'Breastfeeding practices and postpartum weight retention in an Asian cohort' with interest because of the medically (and not only) important issue of post-pregnancy weight gain and the health (and not only) consequences of this gain for both mother and child. Many factors have been identified that determine this gain, whereas the authors of the manuscript have limited themselves to one (but one of the most important), namely the way the baby is fed. This approach to the problematic nature of the study is most reasonable and interesting. The following comments and suggestions are provided according to their relevance to the individual chapters of the manuscript, not according to their perceived importance.

1. Abstract

-        This section should be supplemented with the size of the study group.

-        The division of the subjects into the two study groups analysed (at 6 months postpartum and at 12 months postpartum) should be described precisely but briefly.

-        It would be beneficial to include information in the methodology section indicating that body weight is a value provided by the study participants. This is crucial for maintaining the study's credibility and ensuring that the reader immediately understands the data's origin.

-        Additionally, it may be unnecessary to include the term "public health" in the list of keywords. I believe it would be more appropriate to remove it.

2. Introduction

It is my view that the sentence "In this study, we aimed to examine the association between breastfeeding practices at 6 months and PPWR at 6 and 12 months, using data from the Community-enabled Readiness for first 1000 Days Learning Ecosystem (CRADLE) trial performed in Singapore" (lines 58-61) is unclear in terms of temporality. The phrase "at 6 months and PPWR at 6 and 12 months" is unclear and requires rewording to ensure clarity.

3. Materials and Methods

-        Since the subjects completed the questionnaire online, I think it would be appropriate to add "access to the internet" as another inclusion criterion.

-        Please explain the reason for defining "substantial PPWR" as "as a weight retention of 5 kilograms (kg)". (line 89).

-        In subsection 2.1.2, the authors listed in line 96-97 the confounding factor: "intervention group [14-20]". I don't understand what this factor is - nowhere is there any information about such a division of study subjects. Additionally, what does the inserted reference [14-20] mean?

-        At this point, I will also refer to Table 1, where in the first four rows (on the 5th page of the manuscript) the authors have inserted data on just the "Intervention group" ("control", "Facebook" and "Midwives"). It is imperative that this is clarified as it creates unnecessary confusion.

4. Results

-        It is unclear how individuals were included in the study, as illustrated in Figure 1. In particular, I am uncertain as to the rationale behind the separation of the data set into two blocks following the block labelled "396 Feeding practices data at 6 pp": "379 Weight data at 6 pp" and "347 Weight data at 12 pp". The presentation indicates that the study population was divided into two groups, one studied from birth to six months postpartum and the other from birth to 12 months postpartum. It is my contention that both the figure and the description in the aforementioned section are inaccurate and misleading. Please amend both the figure and the accompanying text to ensure clarity regarding the size of the study group (at the outset and throughout the inclusion process) and the methodology employed.

-        All figures and tables should include an explanation of all abbreviations in the footnotes. This is important because it allows the results to be read without having to find the necessary explanations in the text of the manuscript, thereby greatly improving the perception of the content.

-        There was a mistake in the numbering of chapters 3.2: "3.2 Participant characteristics and substantial PPWR" (line 147) and "3.2 Association between feeding practices and PPWR" (line 168).

-        Please elucidate the sentence "We observed similar associations at 12 months postpartum" (lines 159-160). The lack of data provided makes it challenging to comprehend the content. For instance, analogous data to that presented in Table 1 for "Feeding practices at 6m pp, n (%)" could be included, but only for 12 months.

-        The layout of Table 1 suggests that the study included an analysis of results for two separate groups: the first group 'PPWR at 6m' and the second group 'PPWR at 12m'. However, this is probably not the case. Once more, it is necessary to include a precise description of the study and any breakdown of the study groups.

-        It would have been appropriate to include data on weight changes (in kilograms) in women feeding their babies in three different ways, separated by six months postpartum and 12 months postpartum, in the results section. However, the authors have only included Table 2, which already contains the statistical analysis, in the manuscript. It is important to include this breakdown according to feeding method, as the authors analyse their results from this angle.

-        I am unsure as to the necessity of the passage in lines 139 to 146 of the manuscript text. In this passage, the authors discuss the results in supplementary table 1, which compares the characteristics of participants with complete and incomplete data. Does this contribute to the analysis of the results?

5. Discussion

It is recommended that the authors highlight the following finding of the study: with regard to the method of feeding, the lowest proportion of women breastfed their children compared to women who fed their children in a mixed system (breast milk plus formula) and women who exclusively fed their children with artificial mixtures. This is a highly significant finding in the context of the necessity for action to promote breastfeeding. It is conceivable that the authors may have already provided a description of this in another publication derived from their study. Nevertheless, given that these results were presented in a peer-reviewed manuscript, it is imperative that they be subjected to commentary.

In the sentence: In the sentence "This longitudinal study, utilising data from the CRADLE trial, examined the association between breastfeeding practices and PPWR at 6 and 12 months among women in Singapore" (lines 209-211), it would be beneficial to supplement the time term with further clarification. Could the 6 and 12 months be defined as a reference to the postpartum period? I believe this to be the case and will request an amendment.

6. Conclusions

It is recommended that the full names be used in this section rather than the abbreviations, which apply to both 'PPWR' and 'BMI'.

7. References

The references section comprises 27 items, 16 of which are from the last 10 years. However, a significant proportion of the references are older items, amounting to 11 out of the total.

Furthermore, this section must be aligned with the journal's editorial guidelines. This specifically pertains to the elimination of superfluous capitalisation in manuscript titles and the incorporation of Abbreviated Journal Names.

Round 2

Reviewer 2 Report

Comments and Suggestions for Authors

I thank the authors of the manuscript for their response to my comments in the review form.

I still have a few minor concerns - I will ask you to read them: I have included them under each point and highlighted them in blue.
